# Utilization of In Vivo Imaging System to Study Staphylococcal Sepsis and Septic Arthritis Progression in Mouse Model

**DOI:** 10.3390/pathogens13080652

**Published:** 2024-08-02

**Authors:** Meghshree Deshmukh, Zhicheng Hu, Majd Mohammad, Tao Jin

**Affiliations:** 1Department of Rheumatology and Inflammation Research, Institute of Medicine, Sahlgrenska Academy, University of Gothenburg, 41346 Gothenburg, Sweden; meghshree.vinod.deshmukh@gu.se (M.D.); zhicheng.hu@gu.se (Z.H.); majd.mohammad@rheuma.gu.se (M.M.); 2Center for Clinical Laboratories, The Affiliated Hospital of Guizhou Medical University, Guiyang 550001, China; 3Department of Rheumatology, Sahlgrenska University Hospital, 41345 Gothenburg, Sweden

**Keywords:** IVIS, *S. aureus*, sepsis, septic arthritis, bioluminescent, imaging, mice

## Abstract

*Staphylococcus aureus* [*S. aureus*] is a leading cause of sepsis and septic arthritis, conditions that pose significant medical challenges due to their high mortality and morbidity. No studies have used an in vivo imaging system [IVIS] to monitor *S. aureus* sepsis and septic arthritis. Here, we employed a bioluminescent reporter strain of *S. aureus*, Newman AH5016, administered intravenously to induce sepsis and intra-articularly to induce local septic arthritis in mice. Disease progression was monitored using IVIS to capture bioluminescent signals from kidneys, joints, and whole mice. Cytokines in the blood and joints were measured. The efficacy of cloxacillin treatment was evaluated. In the sepsis model, bioluminescent signals from kidneys, but not from whole mice, were correlated with kidney bacterial load and abscess formation. Ex vivo kidney imaging detected increased bacterial load and abscess formation from day 3 to day 10. Antibiotic treatment significantly reduced kidney signals, correlating with decreased bacterial counts and IL-6 levels, indicating effective infection control. In the local infection model, early-phase bioluminescent signals from joints were correlated with macroscopic arthritis and bacterial burden. Thus, signal detection from kidneys using IVIS is useful for monitoring *S. aureus* sepsis and assessing antibiotic efficacy, though it may only be effective for early-phase monitoring of local septic arthritis.

## 1. Introduction

*Staphylococcus aureus* [*S. aureus*] is the most common causative agent of sepsis and septic arthritis [1]. The hematogenous spread of bacteria often poses a major medical challenge, leading to mortality. Sepsis is a life-threatening condition arising in response to a bacterial infection [2]. *S. aureus* remains one of the leading causes of bacterial sepsis, comparable in incidence and mortality to infections caused by *Escherichia coli*, Streptococcus, and Klebsiella species [3]. Staphylococcal sepsis is a major concern in healthcare due to its high incidence, ranging from 20 to 50 cases per 100,000 population per year, while its significant mortality rate ranges between 10 and 30% [4].

Septic arthritis, on the other hand, is considered the most aggressive joint disease [5]. The incidence of septic arthritis in the general population is approximately 6–10 cases per 100,000 individuals per year [5,6,7,8,9]. Despite prompt treatment, joint damage from septic arthritis often leads to irreversible joint dysfunction in up to 50% of patients [10]. A comprehensive epidemiological study revealed that within 15 years, approximately 9% of septic arthritis patients required arthroplasty, a rate six times higher than the general population [11]. Standard treatments for septic arthritis involve antibiotics and joint drainage [12]. The progress in developing new treatments has stagnated, with current therapies remaining unchanged for 30 years. To mitigate the exaggerated immune response and minimize joint damage, combining antibiotics with immunomodulatory therapies, such as corticosteroids [13] or anti-TNF treatment [14], has been suggested. Nonetheless, these combination therapies pose risks, especially concerning antibiotic resistance [15], and have not been implemented in clinical practice.

Septic arthritis is often caused by hematogenous spread, linking it closely to sepsis. Therefore, preclinical research often studies both diseases concurrently. The interaction between bacteria and the host immune system, particularly innate immunity, is crucial in the onset and progression of septic arthritis and sepsis, as demonstrated by experimental studies [16]. Mouse models for septic arthritis and sepsis, established in the 1980s, have been continuously developed. Initially, these models focused on clinical read-outs such as weight loss, clinical arthritis scores, mortality rates, bacterial kidney counts, and histopathological joint analyses. Since 2017, microcomputed tomography [μCT] scans have replaced histopathological analyses for evaluating joint erosions, offering a less labor-intensive and more precise method.

The in vivo imaging system [IVIS], originally used in oncology research in the early 2000s [17,18,19,20], has since expanded to various fields, including studies on bacterial, viral, and parasitic infections, with a focus on local infections. A subcutaneous catheter-related murine infection model employs IVIS for real-time monitoring of biofilm-associated infection, studying the bioluminescent signal generated by *Staphylococcus epidermidis* in biofilms on medical catheters [21]. Sepsis induced by *Listeria monocytogenes* has been studied to understand the dynamics of host–pathogen interactions, immune responses, and bacterial behavior in real-time imaging techniques [22]. IVIS has been applied in multiple mouse models for *S. aureus* intra-dermal skin infection, epicutaneous skin inflammation, and incisional/excisional wound infections, capturing and analyzing signals from actively metabolizing bacteria [23]. Antibiotic response to bacterial systemic infections has been studied in mouse models using bioluminescent *Listeria monocytogenes*, monitoring bacterial load in the infected organs such as the spleen, bone marrow, intestine, nasopharynx, and brain [24]. Imaging techniques employing bioluminescent mouse cytomegalovirus for infection in the C57BL/6 neonate mouse model [25] and enterovirus reporter constructs for infection in Balb/c mice [26] have shown promise. More advanced techniques like multimodal imaging, which combines signals from both the bioluminescent bacterial reporter strain and fluorescently labeled neutrophils, make the study of host–bacteria interactions more accurate and quantifiable [27].

So far, no studies have used IVIS to monitor the progression of *S. aureus* sepsis and septic arthritis. In this study, our aim was to apply IVIS to assess the severity of *S. aureus* sepsis and septic arthritis. Our objective was to explore how this advanced imaging system can enhance the study of these diseases and validate its efficacy using an infection model treated with antibiotics.

## 2. Materials and Methods

### 2.1. Ethics Statement

All experiments were conducted following the Swedish Research Council’s instructions and regional ethical standards from the Swedish Board of Agriculture’s regulations and recommendations on animal research. Mouse studies [Dnr 5.8.18-02443/2021; Dnr 5.8.18-19171/2019] were reviewed and approved by the regional animal ethics committee [Jordbruksverket] of the University of Gothenburg.

### 2.2. Mice

Female NMRI and C57BL/6 mice, aged 6–8 weeks, were purchased from Envigo [Venray, The Netherlands]. All mice were housed at the animal facility at the University of Gothenburg. Mice were kept under standard temperature and light conditions and were fed laboratory chow and water ad libitum.

### 2.3. Bacterial Strains

The *S. aureus* Newman AH5016 strain, a reporter strain [28] used in experiments, was received from Anschutz Medical Campus, University of Colorado, Aurora, CO, USA. This strain is engineered to express a luciferase gene for bioluminescence. Metabolically active bacteria produce light through the luciferase reaction when excited by specific wavelengths of light, which can be detected using an imaging system. The strain was cultivated for 24 h on horse blood agar plates and then preserved as previously described [29]. Before each experiment, the bacterial solutions were thawed, washed, and adjusted to the desired concentration in sterile phosphate-buffered saline [PBS]. CFUs were counted using serial dilution and plating on horse blood agar plates, followed by incubation and colony counting.

### 2.4. Mouse Models of S. aureus Sepsis and Hematogenous Septic Arthritis

The experimental settings involved systemic infection with *S. aureus*, the reporter strain, for studying and imaging the severity of infection in the mouse models.

For the sepsis experiment, NMRI mice were intravenously [i.v.] injected into the tail vein with 200 μL of *S. aureus* [2.4 × 10^6^ colony-forming units [CFU]/mouse] on day 0. The mice were monitored individually two or three times daily by an observer [M.D.]. The mice were sacrificed on day 3 [*n* = 3] and day 10 [*n* = 14], respectively. The data from three different experiments that were terminated on day 10 were pooled. Weight development and clinical arthritis were evaluated [30] in a blinded manner by two observers [M.D. and Z.H.] throughout the experimental time frame. Mice were anesthetized and imaged for bioluminescent signals in terms of region of interest [ROI], followed by termination for blood, kidneys, and joint collection aseptically collected on day 3 and day 10, respectively. Kidneys collected were imaged for bioluminescent signals ex vivo aseptically. Each kidney was scored and processed individually to ascertain the abscess score and the bacterial load in terms of CFU/mL in each of them on day 3 and day 10 after infection [31]. Signals were also captured from the joints and further analyzed for the extent of bone erosion by μCT. A similar experimental setup was employed to study disease progression in C57BL/6 mice. For this, 200 μL of the *S. aureus* reporter strain [8 × 10^5^ CFU/mouse] was inoculated intravenously into the tail vein of C57BL/6 mice [*n* = 10]. Imaging, blood collection, and joint collection were carried out on day 10 post-infection.

### 2.5. Antibiotic Treatment in Mouse Model of S. aureus-Induced Sepsis

To validate the utility of the in vivo imaging system [IVIS] in a mouse model of *S. aureus* sepsis, NMRI mice were infected with the bioluminescent *S. aureus* reporter strain [2.8 × 10^6^ CFU/mouse]. From day 3 post-infection and onwards, mice were subcutaneously treated with either cloxacillin [10 mg/mouse] in 0.2 mL of PBS twice per day [*n* = 5], known to efficiently eliminate the *S. aureus* infection [14]. The same volume of PBS was given to the control mice [*n* = 5] at the same time points. Upon termination on day 9 post-infection, whole mice were imaged for bioluminescent signals in terms of ROI, blood was collected, and kidneys were removed and imaged for bioluminescent signaling in terms of ROI, ex vivo, aseptically, and processed to evaluate the bacterial load in terms of CFU/mL [31].

### 2.6. Mouse Model of Local S. aureus Septic Arthritis

To study localized infection in knee joints by imaging, NMRI mice [*n* = 16] were sub-grouped into four groups [*n* = 4/group] for day 1, day 3, day 5, and day 9, respectively. Each mouse was intra-articularly [i.a.] injected with 20 μL of *S. aureus* Newman AH5016 strain with a high dose [2.5 × 10^5^ CFU/knee] in one knee or a low dose [2.5 × 10^3^ CFU/knee] in the other knee. Mice were monitored twice daily by an observer [M.D.], and knee size was measured in millimeters [mm] using a caliper on the respective days. Bioluminescent signals were collected in terms of ROI from mouse knee joints and analyzed. Knee joints were collected and processed to quantify the bacterial load in terms of CFU/mL.

### 2.7. Quantification of S. aureus Infection Using In Vivo Imaging System

The mice were anesthetized with an intraperitoneal injection of 200 μL of a ketamine/xylazine mixture and examined by the Newton 7.0 FT500 In Vivo Bioluminescence Imaging System [Vilber Lourmat, Marne-la-Valée, France]. Bioluminescent signals were captured by M.D. from the dorsal side and ventral side of the mice for 5 min each at 37 °C with the sensitivity of ultimate XL [8] using Evolution-Capt software [version 18.16; Vilber Bio Imaging, Marne-la-Valée, France ]. Similarly, bioluminescent signals from the kidney from both sides were captured for 5 min each at 37 °C with the sensitivity of ultimate XL [8]. Kuant software [version 2.5; Vilber Bio Imaging, Marne-la-Valée, France] was employed for image analysis by M.D., wherein a similar scale was applied to all the captured images for further ROI signal estimation. Background ROI was subtracted, and the total count was then measured by selecting either the whole body for mouse imaging or the individual kidney for abscess formation and kidney count. The same ROI is applied to all the images for analysis so as to obtain accurate and optimum signal output.

### 2.8. Homogenate Preparation and Quantification of Bacterial Load in Kidneys and Joints

Kidneys were collected aseptically, and abscess scores were assessed by two investigators [M.D. and Z. H.] in a blinded manner [32]. The scoring system used ranged from 0 to 3 [0 indicates healthy kidneys; 1, 1–2 small abscesses in the kidneys without structure changes; 2, >2 abscesses, but <75% kidney tissue involved; and 3, large amounts of abscess with >75% kidney tissue involved]. The kidneys were then homogenized and plated on horse blood agar plates to quantify the CFU counts.

Knee joints were collected aseptically and homogenized using TissueLyser II [Qiagen, Hilden, Germany]. The homogenate was diluted in sterile PBS, spread on horse blood agar plates, and incubated for 24 h at 37 °C. Viable counts of bacteria were performed, and the bacterial load was quantified as CFUs [33].

### 2.9. Quantification of Immunomodulators Using Enzyme-Linked Immunosorbent Assay

The levels of S100A8/A9 and Interleukin 6 [IL-6] from blood plasma and knee homogenates of NMRI mice infected intravenously with the *S. aureus* reporter strain were analyzed using a DuoSet ELISA kit [R&D Systems, Abingdon, UK] as per the manufacturer’s instructions.

### 2.10. Bone Erosion Estimation Using μCT

Joints were fixed in 4% formaldehyde for 3 days and then transferred to PBS for 24 h. The joints were scanned by SkyScan 1176 μCT [Bruker, Antwerp, Belgium]. The scanning was conducted at 55 kV/455 μA with a 0.2 mm aluminum filter. The exposure time was 47 ms. The X-ray projections were obtained at 0.7° intervals with a scanning angular rotation of 180°. The NRecon software [version 1.6.9.8; Bruker] was used to reconstruct three-dimensional images and evaluated using the CT Analyzer [version 2.7.0; Bruker]. Each joint was evaluated by two researchers [M.D. and T.J.] using a scoring system from 0 to 3 [0: healthy joint; 1: mild bone destruction; 2: moderate bone destruction; and 3: marked bone destruction] as previously described [30].

## 3. Results

### 3.1. In Vivo Imaging Systems Are Able to Follow the Progression and Severity of Sepsis in Mice

To determine the severity of systemic infection in our murine models using the imaging system, NMRI mice were infected with a bioluminescent reporter strain of *S. aureus*, Newman AH5016 [Appendix A], known to induce sepsis and septic arthritis [28]. Mice were monitored to assess the infectious progression by assessing parameters such as weight loss, bacterial load, and abscess formation in the kidneys [16]. A gradual decrease in weight was observed in the infected mice over the period of 10 days [Figure 1A]. There was an increase in both the bacterial load [Figure 1B] and abscess formation [Figure 1C] at the late phase of infection on day 10, compared to the early phase observed on day 3 post-infection. This demonstrates impaired bacterial clearance with an increased severity of the localized kidney infection. In line with clinical data, ROI analysis [Figure 1D] and ex vivo kidney imaging [Figure 1E] also showed a difference between days 3 and 10. No significant difference was observed in the ROI analysis [Figure 1F] with respect to the bioluminescent signals from the whole bodies of live mice on day 3 and day 10 post-infection, respectively [Figure 1G]. We next investigated the efficacy of IVIS in another mouse strain to determine the severity of kidney abscess and infection using ex vivo kidney imaging and whole-body imaging in C57BL/6 mice. A good correlation was observed between the kidney ROI signal and bacterial load [Appendix A] as well as abscess formation [Appendix A], as represented in the image [Appendix A]. Bioluminescent signals from the whole bodies of live mice were also analyzed [Appendix A]. All the above data suggest that ex vivo kidney imaging is a more sensitive method than whole-body imaging to follow the disease progression. IL-6 and S100A8/A9 are closely related to *S. aureus* septic arthritis and sepsis in mouse models. It is known that S100A8/A9 expression levels predict the later development of *S. aureus* septic arthritis [34]. Additionally, IL-6 levels correlate with CT-verified bone erosion scores and kidney bacterial loads in the *S. aureus* septic arthritis model [30]. To understand the dynamic changes of those cytokines during the course of the disease, we analyzed their levels in the blood on days 3 and 10 post-infection. In NMRI mice, the levels of S100A8/A9 were elevated in infected mice on both day 3 and day 10 compared to healthy controls, with no significant difference between day 3 and day 10 [Figure 2A]. In contrast, IL-6 levels peaked on day 3 and significantly decreased by day 10 after infection. The levels were higher in both time points compared to healthy controls [Figure 2B].

### 3.2. Kidney IVIS Signals Correlate Positively with Kidney Abscess Severity and Kidney CFU Counts

The correlation between bioluminescent signals from whole mice or kidneys and various clinical parameters—including weight loss, kidney abscess scores, kidney CFU counts, as well as blood levels of S100A8/A9 and IL-6 analyzed on day 10 post-infection in both NMRI and C57BL/6 mice. Notably, whole-body signals were significantly correlated with weight loss in NMRI mice, whereas the corresponding correlation was absent in C57BL/6 mice. The kidney ROI signals were significantly correlated with both kidney abscess scores and kidney CFU counts across both NMRI and C57BL/6 mice [Table 1].

The whole mouse signal correlated with weight loss and displayed a tendency with regard to S100A8/A9 in NMRI mice, whereas no significant correlation was observed with respect to weight loss, S100A8/A9, IL-6 production, abscess formation, or kidney bacterial load in C57BL/6 mice [Table 1].

### 3.3. The Efficacy of Antibiotic Treatment against Infection Was Monitored Using the Imaging System

To further validate the use of IVIS in the *S. aureus*-induced sepsis model, NMRI mice were inoculated intravenously, with one group receiving cloxacillin treatment. Subsequently, the mice were sacrificed on day 10 after infection. The progression of the infection and treatment efficacy were investigated by assessing weight reduction over the course of the infection. We observed a trend in weight reduction in control mice as compared to cloxacillin-treated mice [Figure 3A]. A reduction in bacterial load was observed in mice treated with cloxacillin as compared to the controls [Figure 3B]. A similar decrease was observed with a diminished ROI signal [Figure 3C]. The whole mouse body imaging [34] and the ROI signal analysis, however, failed to differentiate cloxacillin-treated mice from PBS-treated control mice [Figure 3D]. Additionally, significantly decreased levels of IL-6 [Figure 3E] correlated with the elimination of infection from the body by day 10 post-infection. Bioluminescent signals were captured by ex vivo kidney imaging [Figure 3F] in the cloxacillin-treated mice compared to the controls.

### 3.4. Assessing Clinical Arthritis Progression Using Imaging System

A moderate correlation was found between the signal captured from the affected joints and the bone erosion score [Figure 4A]. No correlation was observed between the joint ROI signal and the clinical arthritis score [Figure 4B] in corraboration with the CT images of the affected joints [Figure 4C]. The bioluminescent signal from representative images of the affected joints has been used for ROI analysis [Figure 4D].

### 3.5. Imaging Technique Can Determine the Infectious Progression over Time in a Local Infection Model

To determine if we can use imaging techniques to study infection in a local mouse model, NMRI mice were intra-articularly infected in the knee joints with the reporter strain, *S. aureus* Newman AH5016, in a high or a 100-fold lower dose. Macroscopic arthritis was assessed until the mice were sacrificed on day 9 post-infection by measuring the mouse knee joint swelling. A significant difference in knee joint measurements was observed between the knee joints injected with high and low doses on day 3 and day 9 [Figure 5A]. Interestingly, a difference in bacterial load was noted on day 3 and day 5 between the high-dose and low-dose groups [Figure 5B]. A significant difference was observed in the ROI signals captured from the mouse knee joints only at the early phase of the disease, on day 3 after infection [Figure 5C], as represented in Figure 5D. S100A8/A9 production showed no significant difference through the whole course of the disease [Figure 5E], whereas a significant difference in IL-6 levels was observed only on day 1 after infection between the groups [Figure 5F].

## 4. Discussion

Our study demonstrates the application of an imaging system in tracking the progression of systemic infection in a murine model, particularly in assessing bacterial load in mouse kidney models. These mouse models have already been evaluated in the past using unique and in-house-developed mouse models appropriate to the experimental setups mentioned here [1,14,15,16,29,30,31,32,33,34,35,36,37]. Additionally, the IVIS system may monitor the progression of septic arthritis, especially in localized septic arthritis models, at the early phase of the disease. A significant advantage of using IVIS in our model systems is the reduction in time and workload. We believe that IVIS has the potential to replace certain conventional methods, such as CFU counts in kidneys.

Recent studies have employed bioluminescent and fluorescent optical imaging and compared them with existing X-ray and μCT imaging to study dynamic changes in bacterial burden, neutrophil recruitment, and bone damage in a mouse orthopedic implant infection model [38]. To our knowledge, no study has used the IVIS system to follow the course of sepsis. Here, we demonstrated that the dynamics of sepsis progression can be traced in a more convenient and non-invasive way by utilizing a bioluminescent reporter strain of *S. aureus* Newman AH5016 and IVIS. Kidney abscess scores and kidney CFU counts are important parameters in a mouse model of *S. aureus* sepsis [16]. Indeed, blood cultures or tissue cultures [livers and spleens] are positive only in the early phase of the disease [3 days post-infection] [39]. Kidney CFU counts are known to correlate with weight loss and infection severity. It is clear that the bacterial load and abscess formation in the kidneys of infected NMRI mice increased from day 3 to day 10. This difference was detected by ex vivo kidney imaging, which captured intense ROI signals on day 3 and day 10. Importantly, a good correlation between kidney ROI signals and kidney CFU counts suggests that ex vivo kidney imaging may replace the time-consuming CFU count methods in future studies. ROI analysis of whole mice did not show a significant difference between day 3 and day 10. This could be attributed to the limited sensitivity in distinguishing the background from the signals in whole mouse body analyses. A future study is planned to improve the sensitivity of whole mouse imaging in order to track the disease progression in real-time.

We have recently shown that gene expression levels of S100A8/A9 in the early stages of *S. aureus* bacteremia predict the later development of septic arthritis in a mouse model [34]. Unlike TNF-alpha, which is usually undetectable on day 10 post-intravenous infection in our mouse model, IL-6 levels remain measurable in the later phase of the disease. IL-6 levels have been positively correlated with the severity of septic arthritis and weight loss [30]. This prompted us to analyze these cytokine levels in the current study to find correlations between cytokine levels and bioluminescent signals, which may represent the severity of infection. Notably, S100A8/A9 levels remained elevated on both day 3 and day 10, while IL-6 levels peaked on day 3 and decreased by day 10. These findings align with previous studies indicating that S100A8/A9 serves as a persistent marker of inflammation [34,40], whereas IL-6 is an early responder [36,37,41] that diminishes as the infection resolves. No significant correlation was found between the blood levels of IL-6 and whole mice body or kidney bioluminescent signals, suggesting that IL-6 levels in the early phase of the disease might be more indicative of disease severity. When comparing different groups of animals in our animal models, such as those receiving different treatments, it is likely more effective to measure IL-6 levels during the early phase. This is because IL-6 levels drop by the later phase of the disease [day 10], making it harder to detect differences. Indeed, our previous findings indicate that differences between the mutant *S. aureus* strain and the parental strain in our model system are more easily detected at earlier time points [42]. However, this needs to be further studied in future research.

The antibiotic treatment options for hard-to-treat multidrug-resistant bacterial infections are limited, resulting in high morbidity and mortality [43]. There is a significant need to develop novel antibiotics and test their efficacy against infections. Our question was whether IVIS could be used to evaluate the efficacy of antibiotics in a mouse model of *S. aureus* sepsis. Indeed, there have already been some studies using non-invasive imaging to study antibiotic efficacy, most often in local infection models, such as *S. aureus* biofilm formation on medical devices in a mouse model [44]. In the current study, cloxacillin treatment significantly reduced the severity of infection in treated NMRI mice. This was evidenced by reduced weight loss, lower bacterial loads, and diminished abscess formation in the kidneys. Additionally, reduced IL-6 levels in treated mice confirmed the successful elimination of the infection. Importantly, whole-body imaging and kidney ROI signal analysis aligned with these clinical findings, showing lower signals in cloxacillin-treated mice compared to untreated controls. Our data demonstrated that ex vivo kidney IVIS is useful for evaluating therapeutic interventions.

In the hematogenous septic arthritis model, we found a significant correlation between imaging signals from affected joints and CT scores, demonstrating the potential of imaging techniques for monitoring arthritis progression. However, the lack of correlation between joint ROI signals and clinical scores suggests that imaging alone may not fully capture the clinical severity of arthritis. This highlights the need for integrating imaging data with clinical assessments to obtain a comprehensive understanding of disease progression [27]. We further studied localized infection progression using intra-articular infection in mice. Despite a 100-fold difference in infection dose, the bacterial counts in joints did not differ on day 9 post-infection. This aligns with the joint ROI signals, which showed differences only in the early phase of the disease, suggesting again that IVIS is a powerful method for measuring the severity of local septic arthritis and the number of infectious organisms at the early phase of the disease. However, despite high CFU counts in the joints, no signal was detected on day 9 post-infection. It is important to note that bioluminant signal generation is more prominent in metabolically active bacteria. We speculate that on day 3, the bacteria are more metabolically active, resulting in a clear signal. By day 9, although the bacterial load has increased, the metabolic activity has decreased, and therefore the signal is no longer detectable.

Overall, our study demonstrates the capability of IVIS to monitor both systemic and localized infections, providing valuable insights into infection dynamics and treatment efficacy. However, in the local joint infection model, the IVIS is not able to detect the living bacteria in the local joints during the late phase of the disease. The whole body imaging is not sensitive enough, possibly due to the high background. Importantly, kidney ROI was significantly correlated with bacterial load in kidneys and kidney abscess formation, suggesting that IVIS has the potential to replace certain conventional methods, such as CFU counts in kidneys.

## Figures and Tables

**Figure 1 pathogens-13-00652-f001:**
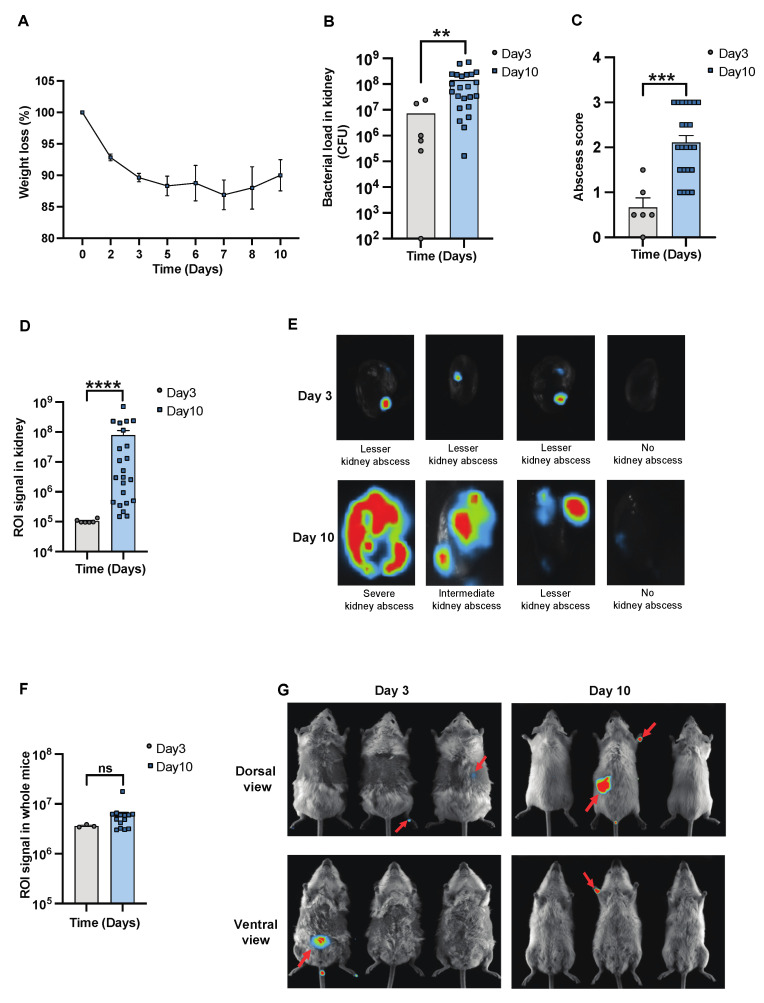
In vivo imaging for assessing the progression and severity of *S. aureus* sepsis. Parameters were assessed in NMRI mice intravenously [i.v.] inoculated with the *S. aureus* Newman AH5016 strain [2.4 × 10^6^ colony-forming units [CFU]/mouse]. The animals were euthanized on days 3 [day 3, *n* = 3] and 10 [day 10, *n* = 14]. [**A**] The percentage changes in body weight up to 10 days post-infection. [**B**] Persistence of *S. aureus* in kidneys [CFU] and [**C**] kidney abscess score. [**D**] Signals were captured and analyzed in terms of region of interest [ROI] from [**E**] ex vivo kidneys, representative images, and [**F**] whole mice body ROI signals, respectively, with [**G**] representative images of whole mice bodies. Arrows indicate bioluminescent signaling. Data for day 10 were pooled from three independent experiments with similar doses. Statistical analyses were performed using the Mann–Whitney U test, with data expressed as the mean ± SEM. ** *p* < 0.01; *** *p* < 0.001; **** *p* < 0.0001; ns = not significant.

**Figure 2 pathogens-13-00652-f002:**
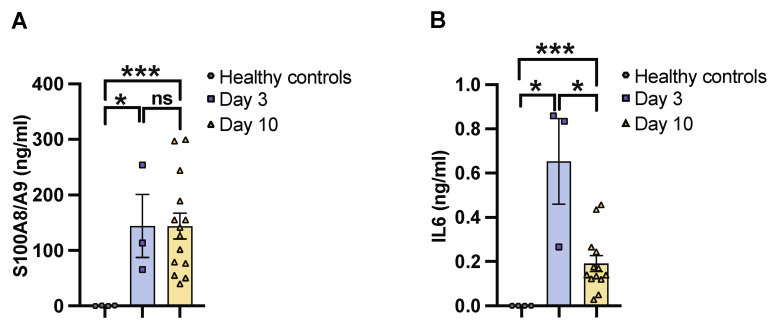
IL6 and S100A8/A9 plasma levels in mice with *S. aureus* sepsis. The levels of [**A**] S100A8/A9 and [**B**] interleukin 6 [IL6] in blood plasma from NMRI mice on days 3 [day 3, *n* = 3] and 10 [day 10, *n* = 14]. The mice were intravenously [i.v.] inoculated with the *S. aureus* Newman AH5016 strain [2.4 × 10^6^ colony-forming units [CFU]/mouse], and healthy controls [*n* = 4] were i.v. injected with phosphate-buffered saline [PBS]. Statistical analyses were performed using the Mann–Whitney U test, with data expressed as the mean ± SEM. * *p* < 0.05; *** *p* < 0.001; ns = not significant.

**Figure 3 pathogens-13-00652-f003:**
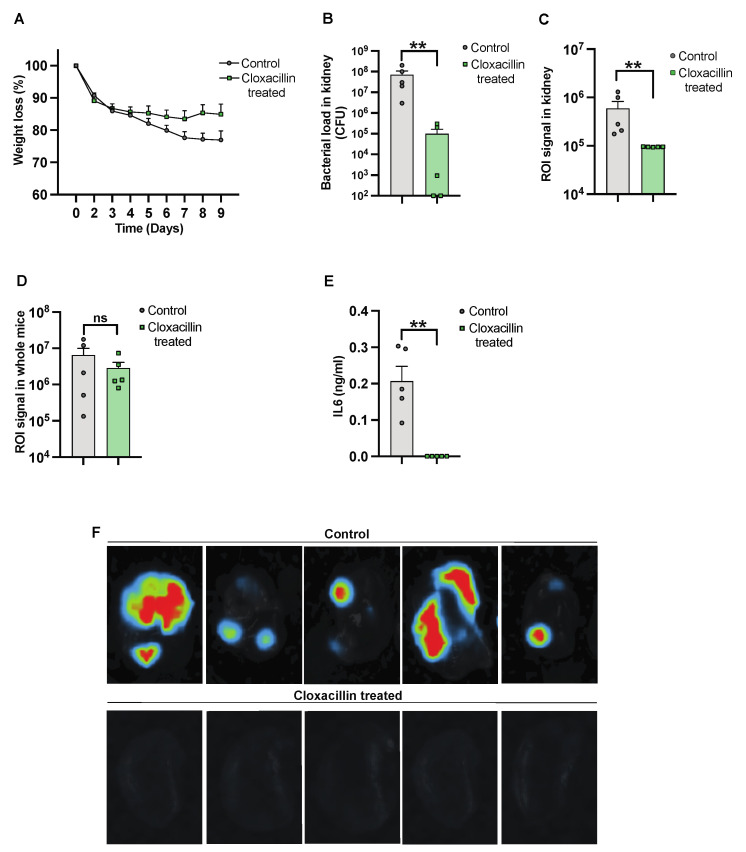
Studying the efficacy of cloxacillin against *S. aureus* infection. NMRI mice were intravenously [i.v.] injected with the *S. aureus* Newman AH5016 strain [2.8 × 10^6^ colony-forming units [CFU]/mouse]. Mice were treated subcutaneously [s.c.] with either cloxacillin [10 mg/mouse, *n* = 5] or PBS [*n* = 5] twice a day from day 3. [**A**] The percentage changes in body weight were monitored up to 9 days post-infection. [**B**] Persistence of *S. aureus* in the kidneys was estimated by quantifying CFU. [**C**] Signals from kidneys were captured and analyzed based on the region of interest [ROI] from ex vivo kidneys. [**D**] The whole mouse body signals were captured and analyzed. [**E**] Interleukin 6 [IL6] levels in blood plasma were determined after the termination of the experiment on day 9 after infection. [**F**] Representative ex vivo images of kidneys from NMRI mice show signals indicating control [PBS-treated] [upper panel] and the effects of cloxacillin-treated [lower panel]. Statistical analyses were performed using the Mann–Whitney U test, with data expressed as the mean ± SEM. ** *p* < 0.01; ns = not significant.

**Figure 4 pathogens-13-00652-f004:**
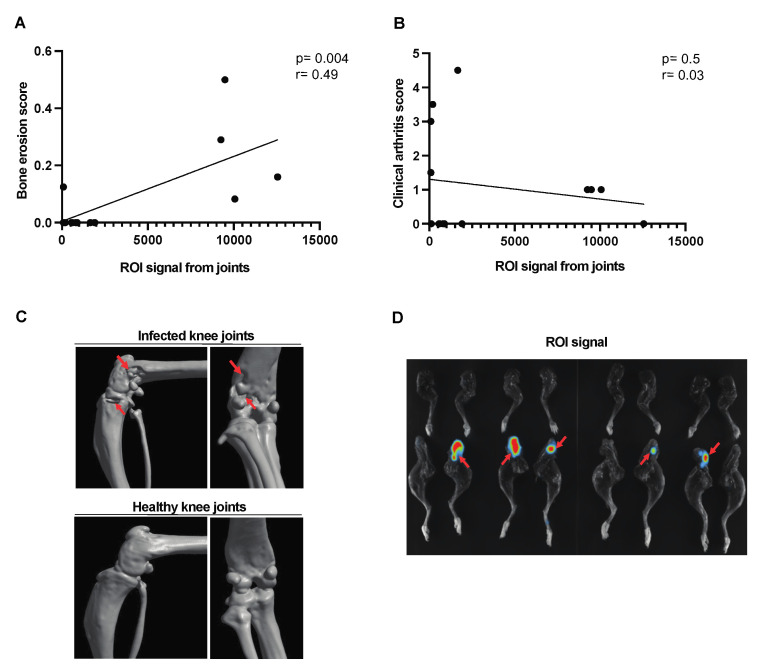
Correlation between the ROI signal from the affected joints and the bone erosion score in mice with hematogenous septic arthritis. NMRI mice intravenously [i.v.] inoculated with the *S. aureus* Newman AH5016 strain [2.4 × 10^6^ colony-forming units [CFU]/mouse] were monitored until day 10 after infection for clinical signs of arthritis and assessed for bone erosion severity based on micro-computed tomography [μCT]. Correlations were drawn between [**A**] bone erosion scores based on the extent of bone erosion visualized by μCT. [**B**] Joint region of interest [ROI] signal and clinical arthritis score assessed clinically. [**C**] Representative computed tomographic images showing bone destruction in infected knee joints [upper panel] and intact healthy knee joints [lower panel]. Arrows indicate bone erosion. [**D**] Representative joint images indicated infection in the affected joints. Arrows indicate bioluminescent signaling. Statistical analyses were performed using simple linear regression to determine the *p*-value and *r*.

**Figure 5 pathogens-13-00652-f005:**
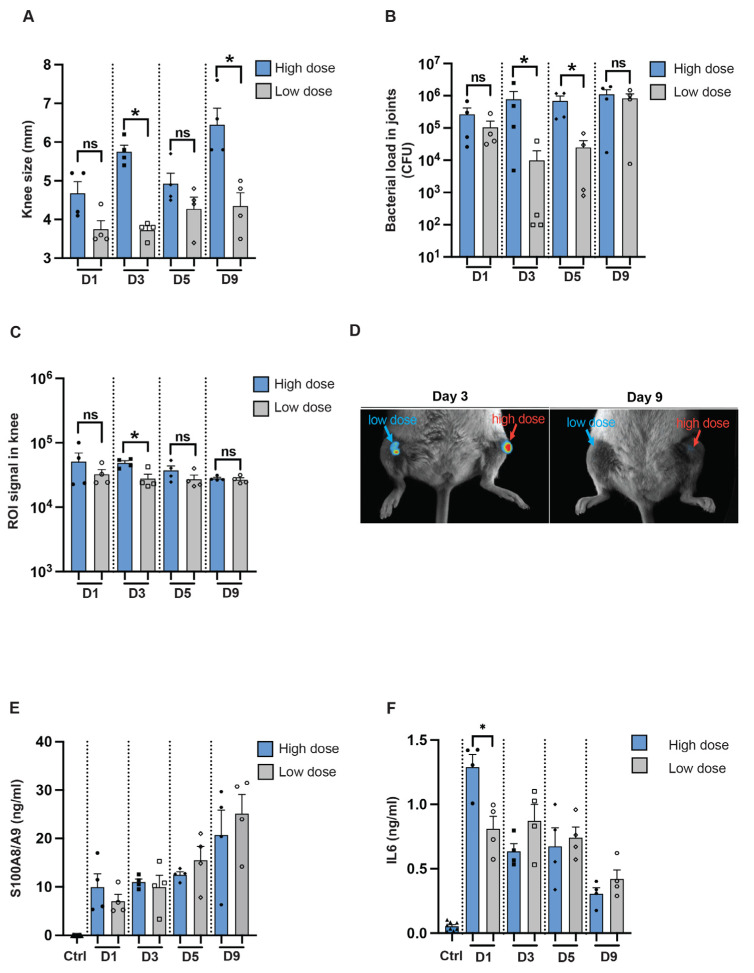
In vivo imaging for assessing infectious progression in a local septic arthritis mouse model. [**A**] Measurement of knee swelling in millimeters [mm] in NMRI mice [*n* = 4/group] on days 1, 3, 5, and 9 after intra-articular [i.a.] injection with 20 μL of *S. aureus* AH5016 strain at a high dose [1.4 × 10^5^ CFU/knee] or low dose [2 × 10^3^ CFU/knee]. [**B**] Bacterial load in the knee joints at the respective time points, expressed as CFU/joint. [**C**] Signals from the knee joints were captured using the Newton imaging system and analyzed in terms of region of interest [ROI] for both high- and low-dose groups. [**D**] Representative images of knee joints (i.a.) injected with high- or low doses of the *S. aureus* AH5016 strain. Arrows indicate bioluminescent signaling. [**E**] S100A8/A9 and [**F**] Interleukin 6 [IL6] levels in knee homogenates on days 1, 3, 5, and 9 post-infection. Statistical analyses were performed using the Mann–Whitney U test, with data expressed as the mean ± SEM. * *p* < 0.05; ns = not significant.

**Table 1 pathogens-13-00652-t001:** Correlation between kidney and whole mouse bioluminescent signal and clinical parameters of sepsis [weight loss, abscess score, kidney count, IL6, and S100A8/A9] from both NMRI and C57BL/6 mice. Statistical analyses were performed using simple linear regression to determine the *p*-value and *r*. ROI = region of interest; ns = not significant.

Parameters	NMRI				C57BL/6			
	Kidney		Whole Mice		Kidney		Whole Mice	
	ROI		ROI		ROI		ROI	
	*p*	*r*	*p*	*r*	*p*	*r*	*p*	*r*
Weight loss	ns	-	0.01	0.3	ns	-	ns	-
Abscess score	0.03	0.18	ns	-	0.0001	0.6	ns	-
KidneyCount	0.008	0.2	ns	-	0.0001	0.6	ns	-
IL6	ns	-	ns	-	ns	-	ns	-
S100A8/A9	0.06	0.2	0.08	0.2	ns	-	ns	-

## Data Availability

The authors declare that the main data supporting the findings of this study are available within the article and its Appendix A. The source data underlying the plots shown in the figures are provided in Appendix A. Extra data are available from the corresponding author upon request.

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
