# Peer review of "Utilization of In Vivo Imaging System to Study Staphylococcal Sepsis and Septic Arthritis Progression in Mouse Model"

_pathogens, 2024, doi:10.3390/pathogens13080652_

Round 1
Reviewer 1 Report
Comments and Suggestions for Authors
This paper illustrates a very nice use of IVIS and bioluminescent S. aureus to track the progression of septic arthritis and sepsis in a mouse model and to assess the efficacy of cloxacillin in its resolution.
This will be a very useful study for the S. aureus and sepsis community. The IVIS in conjunction with uCT is a very powerful tool to illustrate the progression of septic arthritis and the use of cloxacillin in disease resolution and the clear IVIS results are very nice.
The paper is very clearly written.
Table 1 is a bit blurry in my pdf copy so make sure the final version is ok.
Author Response
Response to Reviewer 1:
This paper illustrates a very nice use of IVIS and bioluminescent S. aureus to track the progression of septic arthritis and sepsis in a mouse model and to assess the efficacy of cloxacillin in its resolution.
This will be a very useful study for the S. aureus and sepsis community. The IVIS in conjunction with uCT is a very powerful tool to illustrate the progression of septic arthritis and the use of cloxacillin in disease resolution and the clear IVIS results are very nice.
The paper is very clearly written.
Table 1 is a bit blurry in my pdf copy so make sure the final version is ok.
We greatly appreciate the reviewer's positive feedback on our study and the recognition of its potential impact on the S. aureus and sepsis research community.
Regarding Table 1, we apologize for the blurriness in the provided PDF copy. We will ensure that the final version of the manuscript includes a high-resolution version of Table 1 (line 281) to guarantee clarity and readability. Thank you for bringing this to our attention.
Reviewer 2 Report
Comments and Suggestions for Authors
In this manuscript, the authors examined the utility of IVIS to monitor Staphylococcus aureus infection progression in local and systemic infection models in mice. In a sepsis model, the whole-body ROI signal did not correlate with infection status, but the ex vivo kidney images did. Plasma levels of S100A8/A9 showed no correlation with the infection status, whereas IL6 levels did. When IVIS was used to monitor the efficacy of antibiotics, whole-body images again did not correlate with infection status, but the images of kidneys extracted from the mice did. In the septic arthritis model, the ROI signal from joints showed a weak correlation (r = 0.49) with bone erosion scores but not with clinical arthritis scores (r = 0.03). When bacteria were injected directly into the joints in two different doses (low and high), the ROI signal did not show a strong correlation with either knee sizes or bacterial loads. Based on these results, the authors concluded that "signal detection from kidneys using IVIS is useful for monitoring S. aureus sepsis and assessing antibiotic efficacy."
I have two major concerns about this manuscript. First, I do not agree with the authors’ conclusions. I think this study clearly demonstrates the limitations of IVIS, not its utility. Whole-body images did not correlate with any of the infection models. Only the IVIS images of extracted kidneys showed a correlation with infection status. Since kidney extraction will kill the mice, I do not see an advantage of this method over conventional methods.
The second concern is about the arthritis model in Fig. 5. In the model, the authors appeared to inject two different doses (low and high) of bacteria into the joints of the same mice. In this case, both infection sites will be affected by the same systemic immune responses. I am not sure this is a proper way of conducting the experiment.
Major comments:
2.4. Mouse models..: Here, the authors say that mice were infected on day 3 and 10 and also sacrificed on day 3 and 10.
2.6. Mouse model of local S. aureus septic arthritis: Here, the authors stated that each mouse was infected with a high or a low dose of bacteria. However, Fig. 5D suggests that two different doses of bacteria were injected into the same mouse (Fig. 5D). Please clarify.
Lines 234-235:..measuring IL6 levels during the early phase of the disease is preferable for monitoring the severity of the infection
: Kidney abscess was more pronounce on day 10, but IL6 level decreased, as compared with day 3. Then, I wonder how IL6 level is a good indicator for infection severity.
Lines 291-293: Whole mice body imaging and the ROI signal analysis, revealed a lower signal in cloxacillin-treated mice compared to PBS-treated control mice (Fig. 3D).
: The difference is not statistically significant.
Septic arthritis model and Fig. 5.
: I am not sure whether injecting two different doses into the same mice is a proper way of doing the experiment. The progression of bacterial infection will be determined by the virulence of the infectious agent and the immune response of the host. It is likely that the host immune response might be different between the high and the low dose of bacteria. However, if the same mice are infected with two different doses like here, both infection site will share the same systemic immune responses.
Fig. 5C and 5D: In Fig. 5D, there is almost no signal on Day 9, whereas clear signal is seen on Day 3. However, Fig. 5C shows similar levels of signals for low dose infections.
Fig. 3 and Lines 417 – 418: “Our data demonstrated that IVIS is useful for evaluating therapeutic interventions and tracking their efficacy in real-time.”
: The whole-body ROI signal in Fig. 3D was not correlated with the infection status. Although the kidney images clearly correlate with the infection status, the image was taken for kidney extracted mice. Then, how can this method be useful for tracking drug-efficacy in real-time?
Minor comments
Line 140: ,blood -> , blood
Lines 338 – 339: italicize S. aureus
Author Response
Response to Reviewer 2:
In this manuscript, the authors examined the utility of IVIS to monitor Staphylococcus aureus infection progression in local and systemic infection models in mice. In a sepsis model, the whole-body ROI signal did not correlate with infection status, but the ex vivo kidney images did. Plasma levels of S100A8/A9 showed no correlation with the infection status, whereas IL6 levels did. When IVIS was used to monitor the efficacy of antibiotics, whole-body images again did not correlate with infection status, but the images of kidneys extracted from the mice did. In the septic arthritis model, the ROI signal from joints showed a weak correlation (r = 0.49) with bone erosion scores but not with clinical arthritis scores (r = 0.03). When bacteria were injected directly into the joints in two different doses (low and high), the ROI signal did not show a strong correlation with either knee sizes or bacterial loads. Based on these results, the authors concluded that "signal detection from kidneys using IVIS is useful for monitoring S. aureus sepsis and assessing antibiotic efficacy."
I have two major concerns about this manuscript. First, I do not agree with the authors’ conclusions. I think this study clearly demonstrates the limitations of IVIS, not its utility. Whole-body images did not correlate with any of the infection models. Only the IVIS images of extracted kidneys showed a correlation with infection status. Since kidney extraction will kill the mice, I do not see an advantage of this method over conventional methods.
The second concern is about the arthritis model in Fig. 5. In the model, the authors appeared to inject two different doses (low and high) of bacteria into the joints of the same mice. In this case, both infection sites will be affected by the same systemic immune responses. I am not sure this is a proper way of conducting the experiment.
We highly appreciate your constructive feedback, which has significantly improved this manuscript. Please find our answers to your comments point by point below.
Major comments:
Comment 1: 2.4. Mouse models..: Here, the authors say that mice were infected on day 3 and 10 and also sacrificed on day 3 and 10.
Response 1: We apologize for the unclear statement. The mice were infected on day 0 and sacrificed on day 3 and day 10 respectively. Now we have made the necessary adjustments at line 119- 122 for better clarity in the methods section.
Comment 2: 2.6. Mouse model of local S. aureus septic arthritis: Here, the authors stated that each mouse was infected with a high or a low dose of bacteria. However, Fig. 5D suggests that two different doses of bacteria were injected into the same mouse (Fig. 5D). Please clarify.
Response 2: We apologize for any confusion caused by the description and the representation in Fig. 5D. To clarify, each mouse was injected with both a high dose and a low dose of bacteria in different joints to compare the local immune response within the same animal. This approach was designed to minimize the use of animals in the study, adhering to the 3R principles of replacement, reduction, and refinement in animal ethics. Additionally, comparing the immune responses at two different sites within the same mouse increases the statistical power of the study. Most importantly, our previous studies have suggested that local S. aureus in mouse joints does not metastasize to the blood or other organs in immunocompetent mice.
Comment 3: Lines 234-235:..measuring IL6 levels during the early phase of the disease is preferable for monitoring the severity of the infection
: Kidney abscess was more pronounce on day 10, but IL6 level decreased, as compared with day 3. Then, I wonder how IL6 level is a good indicator for infection severity.
Response 3: It is a very good point, and we fully agree with you. Our intention was to emphasize that when comparing different groups of animals, such as those receiving different treatments, it is likely more effective to measure IL-6 levels during the early phase. This is because IL-6 levels drop by the later phase (day 10), making it harder to detect differences. Indeed, our previous findings indicate that differences between the mutant S. aureus strain and the parental strain in our model system are more easily detected at earlier time points(1). We have now omitted the sentence at line 251-254.
Comment 4: Lines 291-293: Whole mice body imaging and the ROI signal analysis, revealed a lower signal in cloxacillin-treated mice compared to PBS-treated control mice (Fig. 3D). The difference is not statistically significant.
Response 4: Thank you for your insightful examination of our manuscript. We fully agree that there is a limitation in our experimental setup regarding whole-body imaging of mice, and the observed difference is not statistically significant. We have rephrased the sentence to reflect this accurately at lines 310-312.
Comment 5: Septic arthritis model and Fig. 5.
: I am not sure whether injecting two different doses into the same mice is a proper way of doing the experiment. The progression of bacterial infection will be determined by the virulence of the infectious agent and the immune response of the host. It is likely that the host immune response might be different between the high and the low dose of bacteria. However, if the same mice are infected with two different doses like here, both infection site will share the same systemic immune responses.
Response 5: We agree that both infection sites share the same systemic immune responses. However, in our local septic arthritis model, the local immune responses are the most predominant compared to systemic immune responses, as local S. aureus in mouse joints does not metastasize to the blood or other organs in immunocompetent mice. This strategy has been employed even in the S. aureus components-induced local arthritis model, allowing for the detection of severity differences in local arthritis between different knees in the same mice(2). As mentioned earlier, our intention was to minimize animal use and increase statistical power. We hope you agree with us.
Comment 6: Fig. 5C and 5D: In Fig. 5D, there is almost no signal on Day 9, whereas clear signal is seen on Day 3. However, Fig. 5C shows similar levels of signals for low dose infections.
Response 6: Thank you for your careful examination. Figure 5D is a representative figure showing a weak signal from a low dose joint on day 3. You may notice in Figure 5C that one knee had a higher signal compared to others and to the knees on day 9.
It is important to note that signal generation is more prominent in metabolically active bacteria. On day 3, the bacteria are more metabolically active, resulting in a clear signal. By day 9, although the bacterial load has increased, the metabolic activity has decreased, and therefore the signal is no longer detectable. We have added this point into the discussion in lines 458 - 463.
Comment 7: Fig. 3 and Lines 417 – 418: “Our data demonstrated that IVIS is useful for evaluating therapeutic interventions and tracking their efficacy in real-time.”
: The whole-body ROI signal in Fig. 3D was not correlated with the infection status. Although the kidney images clearly correlate with the infection status, the image was taken for kidney extracted mice. Then, how can this method be useful for tracking drug-efficacy in real-time?
Response 7: Thank you so much for constructive comments. We fully agree with you. Now we have revised our conclusion in both abstract and discussion to emphasize the limitation of IVIS system for non-invasive follow up of our model system in real time. We now made the necessary adjustments in the statement in lines 417-418.
Minor comments
Comment 8: Line 140: ,blood -> , blood
Response 8: Thank you for pointing this out. We have now made the necessary correction in line 140.
Comment 9: Lines 338 – 339: italicize S. aureus
Response 9: Thank you for pointing this out. We have now made the necessary correction in line 338 - 339.
References:
- Hu Z, Kopparapu PK, Ebner P, Mohammad M, Lind S, Jarneborn A, et al. Phenol-soluble modulin α and β display divergent roles in mice with staphylococcal septic arthritis. Commun Biol. 2022;5(1):910.
- Mohammad M, Nguyen MT, Engdahl C, Na M, Jarneborn A, Hu Z, et al. The YIN and YANG of lipoproteins in developing and preventing infectious arthritis by Staphylococcus aureus. PLoS Pathog. 2019;15(6):e1007877.
Reviewer 3 Report
Comments and Suggestions for Authors
This manuscript describes the application and usefulness of IVIS to monitor S. aureus sepsis and septic arthritis. This is highly interesting study and expand the in vivo experimental methods to scientific community. This reviewer comments following points and questions to improve the original version.
1. For readers who are not familiar with this methodology, it is recommended to explain S. aureus Newman AH5016 strain briefly. In method section 2.3, this strain was just mentioned with reference, but it is no sufficient. The points of mechanisms relevant to IVIS should be described. In Results, 3.1.1, line 208-209, this repeated description is not necessary.
2. According to Methods, bacterial strains were stored in freezer, and thawed and washed, and adjusted cell number. In such case, did authors cultured the strain on liquid or agar media? Were the bacterial cells in the logarithmic proliferation phase used for experiments? How was CFU counted and adjusted for the inoculating strain? Line 176-179 shows CFU count for samples after infection in mice. Please note that accuracy of CFU, and viability of bacterial cells may be easily changed by minor conditions in experiment.
3. Author injected 200ul of bacterial strain. In which liquid media or solution were the bacterial cells mixed?
4. Why did authors select IL6 and S100A8/A9 as immunomodulators? What is the meaning of difference in the changing levels of them in mice infected with SA?
Author Response
Response to Reviewer 3:
This manuscript describes the application and usefulness of IVIS to monitor S. aureus sepsis and septic arthritis. This is highly interesting study and expand the in vivo experimental methods to scientific community. This reviewer comments following points and questions to improve the original version.
Comment 1: For readers who are not familiar with this methodology, it is recommended to explain S. aureus Newman AH5016 strain briefly. In method section 2.3, this strain was just mentioned with reference, but it is no sufficient. The points of mechanisms relevant to IVIS should be described. In Results, 3.1.1, line 208-209, this repeated description is not necessary.
Response 1: We appreciate the reviewer’s suggestion to provide more context on the S. aureus Newman AH5016 strain. We have revised the methods section to include a brief explanation of this strain, in lines 107 – line 110, highlighting its relevance and mechanisms pertinent to IVIS. Specifically, we will describe its bioluminescent properties, which allow for real-time tracking of infection.
Comment 2: According to Methods, bacterial strains were stored in freezer, and thawed and washed, and adjusted cell number. In such case, did authors cultured the strain on liquid or agar media? Were the bacterial cells in the logarithmic proliferation phase used for experiments? How was CFU counted and adjusted for the inoculating strain? Line 176-179 shows CFU count for samples after infection in mice. Please note that accuracy of CFU, and viability of bacterial cells may be easily changed by minor conditions in experiment.
Response 2: We apologize for the lack of detail in the methodology regarding the preparation of bacterial cultures. We cultured the S. aureus Newman AH5016 strain for 24 hours on horse blood agar plates and then preserved in freezing medium. Before each experiment, the bacterial solutions were thawed, washed, and adjusted to the desired concentration in sterile PBS. CFUs were counted using serial dilution and plating on horse blood agar plates, followed by incubation and colony counting. We have included these details in the revised methods section, lines 112 – lines 114, to clarify the preparation and accuracy of bacterial inocula.
Comment 3: Author injected 200ul of bacterial strain. In which liquid media or solution were the bacterial cells mixed?
Response 3: The bacterial cells were suspended in phosphate-buffered saline (PBS) for injection. We have included this information in the methods section, line 112 - 113 to provide clarity on the preparation of the bacterial suspension used for inoculation.
Comment 4: Why did authors select IL6 and S100A8/A9 as immunomodulators? What is the meaning of difference in the changing levels of them in mice infected with SA?
Response 4: We selected IL-6 and S100A8/A9 as immunomodulators due to their critical roles in the immune response to bacterial infections. IL-6 orchestrates the systemic response to infection, including fever and the activation of adaptive immunity, while S100A8/A9 provides direct antimicrobial actions and aids in the recruitment of immune cells to the infection site(1, 2). Importantly, both molecules are closely related to S. aureus septic arthritis and sepsis in mouse models. It is known that S100A8/A9 expression levels predict the later development of S. aureus septic arthritis(3). Additionally, IL-6 levels correlate with CT-verified bone erosion scores and kidney bacterial loads in the S. aureus septic arthritis model(4). We have now included this information alongwith the references in lines 240- lines 245, in the results section for clarity.
References:
- Hunter CA, Jones SA. IL-6 as a keystone cytokine in health and disease. Nat Immunol. 2015;16(5):448-57.
- Ehrchen JM, Sunderkötter C, Foell D, Vogl T, Roth J. The endogenous Toll-like receptor 4 agonist S100A8/S100A9 (calprotectin) as innate amplifier of infection, autoimmunity, and cancer. J Leukoc Biol. 2009;86(3):557-66.
- Deshmukh M, Subhash S, Hu Z, Mohammad M, Jarneborn A, Pullerits R, et al. Gene expression of S100a8/a9 predicts Staphylococcus aureus-induced septic arthritis in mice. Front Microbiol. 2023;14:1146694.
- Fatima F, Fei Y, Ali A, Mohammad M, Erlandsson MC, Bokarewa MI, et al. Radiological features of experimental staphylococcal septic arthritis by micro computed tomography scan. PLoS One. 2017;12(2):e0171222.
Round 2
Reviewer 2 Report
Comments and Suggestions for Authors
In this revised manuscript, the authors properly addressed most of my previous concerns.
However, there are still some minor concerns described below:
Line 122: respectively.. -> respectively.
Line 226: ].. -> ].
Make sure to italicize all bacterial names throughout the manuscript (e.g. line 227)
Line 228: Referencing is not in the correct format.
Line 287: ..[Fig 3B] similar -> ..[Fig 3B]. A similar
Line 293: It would be better to add what the ex-vivo imaging results mean in terms of the utility of IVIS.
Line 312: A strong correlation..
: For r=0.49, I guess a “moderate” correlation will be more appropriate.
Line 421: successful elimination of the infection: cloxacilline treatment did not eliminate the infection (Fig 3B and C).
Comments on the Quality of English Language
Except for some grammatical errors, this manuscript is written clearly.
Author Response
Thank you for your thorough review and valuable comments on our manuscript. We appreciate the time and effort you have invested in providing your feedback.
We have carefully considered all the comments and have made the necessary revisions to the manuscript. Please find below response to each of your comments:
However, there are still some minor concerns described below:
Comment 1: Line 122: respectively.. -> respectively.
Thank you for pointing this out. We have now made the necessary correction in line 122.
Comment 2: Line 226: ].. -> ].
Thank you for pointing this out. We have corrected the punctuation error in line 226.
Comment 3: Make sure to italicize all bacterial names throughout the manuscript (e.g. line 227)
We have reviewed the manuscript and ensured that all bacterial names are italicized, including the instance on line 217
Comment 4: Line 228: Referencing is not in the correct format.
We have corrected the referencing format in line 228 to ensure consistency with the journal’s guidelines.
Comment 5: Line 287: ..[Fig 3B] similar -> ..[Fig 3B]. A similar
We have revised the sentence for clarity. The updated text now reads “..[Fig 3B]. A similar.” This change can be found in line 310 of the revised manuscript.
Comment 6: Line 293: It would be better to add what the ex-vivo imaging results mean in terms of the utility of IVIS.
We carefully reviewed the manuscript and were unable to locate the text or issue mentioned in your comment. Could you please provide more details or specify the exact location of this comment? We want to ensure that we address all your concerns appropriately.
Comment 7: Line 312: A strong correlation..
: For r=0.49, I guess a “moderate” correlation will be more appropriate.
We have revised the description of the correlation from “strong” to “moderate.” The updated text now reads: “A moderate correlation..” This change can be found in line 335 of the revised manuscript.
Comment 8: Line 421: successful elimination of the infection
: cloxacilline treatment did not eliminate the infection (Fig 3B and C).
We carefully reviewed the manuscript and were unable to locate the text or issue mentioned in your comment. Could you please provide more details or specify the exact location of this comment? We want to ensure that we address all your concerns appropriately.